# The Impact of a Grocery Store Closure in One Rural Highly Obese Appalachian Community on Shopping Behavior and Dietary Intake

**DOI:** 10.3390/ijerph19063506

**Published:** 2022-03-16

**Authors:** Rachel Gillespie, Emily DeWitt, Stacey Slone, Kathryn Cardarelli, Alison Gustafson

**Affiliations:** 1Family and Consumer Sciences Extension, College of Agriculture, Food & Environment, University of Kentucky, Lexington, KY 40506, USA; emily.dewitt@uky.edu; 2Dr. Bing Zhang Department of Statistics, College of Arts and Sciences, University of Kentucky, Lexington, KY 40506, USA; stacey.slone@uky.edu; 3Department of Health, Behavior & Society, College of Public Health, University of Kentucky, Lexington, KY 40506, USA; kathryn.cardarelli@uky.edu; 4Department of Dietetics and Human Nutrition, College of Agriculture, Food & Environment, University of Kentucky, Lexington, KY 40506, USA; alison.gustafson@uky.edu; 5College of Nursing, University of Kentucky, Lexington, KY 40536, USA

**Keywords:** food environment, rural, obesity, grocery shopping, behavior change

## Abstract

Research has examined how the entry of grocery stores into neighborhoods influences dietary outcomes, yet limited evidence suggests a direct correlation between opening a store and changes in dietary intake. A factor that might influence individuals’ behavior more directly is the closing of a grocery store where residents shop. This study aims to examine how a grocery closure in a rural Appalachian high poverty county is associated with dietary intake. A cohort of *n* = 152 individuals were recruited to participate in a longitudinal study examining purchasing habits and dietary intake. At time point two, one year later, *n* = 74 individuals completed the survey via phone. Results indicate those that switched from shopping at a local grocery store to a supercenter significantly increased their dietary intake of fruit (0.2 ± 0.8), fruits and vegetables (1.4 ± 2.7), alcohol (grams) (17.3 ± 54.1), and tomato sauce (0.1 ± 0.3). A local grocery store closure was associated with a change in shopping behavior and dietary intake. Community-level interventions targeting dietary behaviors must account for neighborhood food environment influences, including grocery store availability. Policy aimed at improving food access in rural communities need to consider approaches to improving a variety of food venues with affordable healthy food, while addressing the evolving grocery shopping behaviors of consumers.

## 1. Introduction

Improving dietary intake among rural and lower-income populations is a national priority. At the individual level of dietary intake, a host of proximal and distal factors influence purchasing habits and thus what is consumed [1,2,3]. Specifically, among lower-income and rural communities, there is strong evidence to suggest that these neighborhoods often lack access to supercenters which constrains residents’ purchasing ability of affordable healthier food items [4]. The United States Department of Agriculture Economic Research Center (USDA ERS) defines supercenters as food retail outlets in which a general line of groceries, in combination with a general line of merchandise such as apparel, furniture, and appliances are offered and available for purchase [5]. In contrast, grocery stores or supermarkets are food retail outlets offering a general line of food, such as fresh, canned, or frozen foods including prepared meats, fish, and poultry [5]. It was recently reported that the number of grocery stores and convenience stores, which are most commonly found in rural communities, have been steadily declining since 2009 [6]. Furthermore, the median number of grocery stores have decreased by nearly 40% among both rural and nonmetro counties, which has long reaching impacts on food availability, dietary intake, and food security overall among these populations and communities [6]. 

Recent efforts in urban communities have targeted opening grocery stores in “food deserts” (i.e., neighborhoods lacking grocery and supercenters). For example, the Healthy Food Financing Initiative (HFFI) policy has invested more than $500 million in assistance to bring grocery stores to underserved markets in urban and rural communities [7]. Recent evaluation of these efforts in urban areas found that perception of residents for healthy food access did improve [8]. Yet, when assessing actual dietary change or overall availability of healthy food, the introduction of a neighborhood grocery store did little to improve individual healthy food intake or overall healthfulness of the neighborhood [8,9,10]. In addition, to date, there has been little to no examination of the exit of a grocery store in low-resource, high poverty, rural communities [11]. 

Few studies to date have examined the impact on the closure of a grocery store in a rural community on changes in shopping behavior, as current research generally assesses the impact of the opening of food outlets rather than the latter [9]. The community in which this study took place is located within a rural highly obese (i.e., adult obesity prevalence greater than 40%) Appalachian community in which food outlets and healthy food accessibility are scarce and which suffered the closure of one of the only local grocery stores frequented by county residents in January 2020. Our study provides important perspective by providing evidence of the impact of rural grocery store closures on dietary intake. Although randomized controlled trials (RCTs) remain the gold standard to assessing causal impacts due to a treatment or exposure, natural experiments, such as this, serve as valuable impact evaluations within food and retail settings as RCTs are not always feasible to test for changes in behavioral and health outcomes [12]. Therefore, this study was a natural experiment to assess how the closure of a grocery store impacts food outlet shopping choice, dietary intake, and purchasing patterns among a cohort sample over the course of one year.

## 2. Materials and Methods

### 2.1. Study Setting

As part of a larger multi-year High Obesity Program (HOP) project funded by the Centers for Disease Control and Prevention (CDC), the current study addresses access to nutritious foods among a rural, low-income population in Eastern Kentucky. Martin County, Kentucky, is in the Central Appalachian region of the U.S. and has an adult obesity prevalence greater than 40%. In addition to high obesity prevalence, the county experiences high rates of cancer prevalence and other obesity-related chronic illnesses such as Type II diabetes and heart disease [13,14]. The county also faces persistent poverty [15], with a median household income of approximately $30,320, and 32% of residents currently living in poverty [16]. One in five adults are considered food insecure [17], with 22% participating in the Supplemental Nutrition Assistance Program (SNAP) [18]. 

Furthermore, the food environment within this community is sparce and offers poor healthy food access to its residents. At the initiation of this cohort study, when the baseline data collection occurred in the fall of 2019, there were three local grocery stores available in the community. There were no supercenters within the county, and all other grocery shopping access required traveling outside county lines. This is consistent with previous studies in which individuals within the Appalachian region report traveling on average 10 miles one way to the closest grocery store [19]. However, in January 2020, one of the three grocery stores in the county suddenly closed, leaving only two available local grocery stores. The county serves a population of 11,287 as of 2020 and is 229 square miles [20]. These two stores are 10 miles apart and serve opposite ends of the county in each of the two incorporated towns, further limiting residents to a handful of other nontraditional food outlets such as convenience stores, gas stations, or discount stores. Figure 1 depicts a county map with available food outlets to residents identified.

### 2.2. Survey Administration

A prospective cohort was enrolled in the fall of 2019. Methodology and findings from time point one can be found elsewhere [21]. Inclusion criteria were consistent with time point one. Survey participants had completed the survey at time point one and were 21 years of age or older, had lived in the county for at least one year, spoke English, reported no plans to move from the county within the next three years, and had not been diagnosed with cancer. The survey instrument aimed to assess dietary intake, food shopping behaviors, physical activity levels, and community perspectives among community residents. All initially enrolled cohort participants (*n* = 152) were sent a mailed post card and contacted via phone in the fall of 2020 with an invitation to participate in the second time point of data collection. Research personnel attempted to call participants three times before designating them as “no contact”. Figure 2 outlines the time point two enrollment processes. Of the 152 participants enrolled at baseline, 85 agreed to participate in the follow-up survey, and 67 individuals did not participate. Of the 67 who did not participate, 46 were unable to be reached after three phone call attempts and mailed postcard, 17 declined to participate, 1 was deceased, and 3 were no longer reachable at the phone number provided. Of the 85 who planned to participate, there were 74 cohort participants who completed the survey for the second time point.

The surveys were verbally administered via telephone by trained researchers in the fall of 2020 (October and November) and winter of 2021 (January). The administration took approximately 30–45 min to complete. All participants received a $25 incentive to be used at a local gas station as compensation for their time. The University of Kentucky Institutional Review Board (IRB) approved the research design and all study materials (protocol 48905). 

### 2.3. Survey Measures

The survey instrument utilized for this cohort study was consistent between baseline and time point two with the addition of new questions. The assessment measures presented within this manuscript were consistent between the two time points and were collected, recorded, and analyzed with only income being statistically significant different, with participants reporting higher income at baseline more likely to participate at time point two, *p* = 0.03.

#### 2.3.1. Grocery Shopping Measures

Questions were adapted from the United States Bureau of Labor Statistics American Time Use Survey Eating & Health Module to assess grocery shopping preferences among the cohort [22]. Participants were asked to respond to the question “Where do you get most of your groceries?” and were provided five responses including a grocery store, supercenter, discount store, gas station or corner store, or other location with multiple selections allowed. Once respondents indicated where they purchased most of their groceries, participants were asked to indicate what the primary reason(s) they shopped there was. Reasons for shopping at the selected food venue included price, location, variety, or quality. These questions were assessed at both time points among the cohort study participants.

#### 2.3.2. Dietary Intake Assessment

Dietary intake was assessed utilizing the National Cancer Institute (NCI) Fruit and Vegetable Screener [23,24]. The NCI screener allows participants to self-report usual intake of a variety of fruit and vegetable items from never to ≥five times per day. There is a breadth of data linking the effect of fruit and vegetable intake to disease prevention [25,26,27], notably, cancer, heart disease, and stroke. In addition, there is a body of research linking excessive sugar sweetened beverages (SSB) consumption with prediction of obesity and subsequent health outcomes. Thus, for purposes of disease prevention, these vital dietary intake variables were selected. Respondents also self-reported portion sizes for each item they report consuming. 

The Beverage Intake Questionnaire (BEVQ-15), a validated instrument, was used to assess a variety of beverage intakes among participants [28,29]. Frequency of consumption during the last month, and amount consumed at each occurrence, was collected to determine intake amounts. In order to determine SSB calories and grams, the following sugar containing beverages from the questionnaire were combined—sweetened juices, regular soda, energy drinks, sweet tea, and coffee with cream and sugar. This methodology has been used previously to determine consumption habits and amounts among a study sample [30]. Additionally, all milk categories—regular, reduced-fat, skim, and milk alternatives—were combined to determine grams consumed, as well as all forms of alcohol—beer, wine, and hard liquor—to determine grams consumed. 

The assessment in dietary habits was calculated by measuring three different associations. “No Change” was defined as the those who did not purchase most of their groceries at a supercenter at either time point; “Started” was defined as individuals that reported purchasing most of their groceries at a supercenter at time point two, but not at time point one; and “Already” defined those who purchased most of their groceries at a supercenter at time point one. 

### 2.4. Analysis

Data were entered directly into REDCap (Vanderbilt University, Nashville, TN, USA) by trained researchers during survey administration. Data were exported and analyzed using SAS 9.4 (SAS Institute, Cary, NC, USA). Medians, ranges, counts, and percentages were calculated for groups as appropriate. Comparison between responders and non-responders at time point two were calculated using Wilcoxon tests for age and Chi-square and Fisher’s exact tests for all other demographic variables, as appropriate. 

McNemar’s tests were conducted to determine statistically significant changes in shopping behaviors between time point one and time point two. The changes in dietary habits were assessed for normality and were compared across the three food retail shopping categories with a one-way analysis of variance. However, income was assessed in and out of models and no changes to point estimates were observed, so models were retained without. Significance was set at *p* < 0.05 for all analyses. 

## 3. Results

### 3.1. Study Sample

Demographic characteristics of this sample can be found in Table 1 compared to characteristics among this sample reported at baseline. Participants (*n* = 74) at time point two had a mean age of 59.5 years, with more than two-thirds identifying as female (*n* = 51, 68.9%), which is generally reflective of the overall county make up and population; however, proportionately more females participated in the cohort survey than indicated living in the county [31]. Most participants reported an annual household income less than $20,000 (*n* = 44, 61.1%), with 41.9% receiving SNAP benefits in the last month. Travel time to the reported grocery store was not assessed at baseline; however, it was collected at time point two among participants. The majority (*n* = 24, 32.4%) reported traveling between 20 and 30 min one way to get to their food shopping destination. Less than 10 min (*n* = 19, 25.7%) followed by more than 30 min of travel time (*n* = 17, 23%) were the next most commonly reported. 

### 3.2. Grocery Shopping Habits

As seen in Table 1, grocery shopping habits indicate that participants purchased most of their groceries at either a grocery store (63.5%) or supercenter (48.6%). The primary reasons for shopping these venues were location (41.9%) and price of food items (29.7%). Furthermore, time point two descriptive statistics revealed location became the primary motivator for choosing to shop at the reported grocery venue (41.9%) at time point two (40.8% at time point one). Price was the predominant motivator at time point one (42.1%), while only 29.2% of respondents indicated price as the reason for selecting a store at time point two.

In order to assess change in habits and motivations over time, Table 2 compares the change in behaviors from time point one to time point two among the same cohort participants (*n* = 74) when asked where they get most of their groceries and the primary reason for shopping there. Compared to time point one, 56.8% of individuals reported buying most of their groceries at a grocery store at both time point one and time point two. Most significantly, 27.0% of individuals reported purchasing most of their groceries at a supercenter at time point two when they did not at time point one (*p* = 0.0004). Not everyone who stopped shopping at a grocery store started shopping at a supercenter, but many did.

Of note and most significantly when comparing shopping motivations from time point one to time point two, 25.7% of respondents indicated variety was the primary reason for their selected shopping store, whereas at time point one only 12.2% shared that this was prioritized (*p* = 0.01). This was the only observed significant change among the cohort from time point one to time point two, whereas the other motivations were not significant. 

### 3.3. Dietary Intake

Table 3 reports the overall mean fruit and vegetable, and beverage intake changes from time point one to follow-up at time point two among those who reported changing the venue from which they purchased most of their groceries from a grocery store to a supercenter. For those that switched from shopping at a grocery store to a supercenter, their fruit intake increased by 0.1 (SD = 0.6), tomato sauce intake increased by 0.1 (SD = 0.3), overall fruit and vegetable consumption increased by 1.4 (SD = 2.7), and grams of alcohol increased by 17.3 (SD = 54.1). This is significantly more than those who had no change in shopping habits or already bought most of their groceries at supercenters and experienced no change or decreases in average intake for all four dietary measures. 

## 4. Discussion

Rural American consumers continue to face an array of barriers to healthy eating that have not been ephemeral, due in large part to a lack of healthy food options available to them. Furthermore, supercenters have grown 526% since 1990 until 2015, while the presence of grocery stores has been decreasing by 36% in the number of establishments in rural or nonmetro counties across the U.S. [6]. These shifts in the retail food environment continue to influence food shopping preferences among consumers as evidenced by these findings.

This study depicts the fluctuation in grocery shopping behaviors as a result of a diminishing food access environment in a rural Appalachian community. A larger percentage of participants reported shopping at supercenters at follow-up compared to time point one. As more consumers were forced to frequent other grocery shopping sources, such as a supercenter, variety of items became a more significant reason for selecting this venue. We postulate two main motives for this change in preferred grocery shopping venue. First, the closure of one grocery store in the county seat between data collection time points may have influenced shopping behaviors of participants, further limiting proximal access to affordable and nutritious food. Previous efforts have been made to support food outlet fixtures in rural food environments [32]; however, rural communities continue to face limited access to food retailers and struggle to support larger food outlets, such as supercenters or grocery stores. The closure of this store is one factor that led to change in shopping practices and thus dietary intake. Previous research has indicated that store closures are a result of economic distress, in which rural communities have faced disproportionately more than their urban counterparts over the last several decades [6]. These communities are experiencing a variety of impediments that collectively are contributing to the destituteness, such as higher poverty rates and decreasing populations [33,34], thereby contributing to loss of retail businesses such as grocery stores. Additionally, grocery stores in rural communities more frequently are independently owned, rather than affiliated with a national retail supercenter [35,36]. This further reinforces the implausible potential for rural grocery stores to weather long-term economic distress in their community, while experiencing frangible pressure that chain stores may not. 

Second, the COVID-19 pandemic may have further contributed to the change in food shopping venues among our sample, particularly with “location” being the primary reason for shopping at chosen venues. The COVID-19 pandemic influenced many aspects of the food environment and affected how individuals access and procure food [37]. Online ordering became more widely available and more popular among consumers in response to the COVID-19 pandemic [38]. Thus, some of our participants may have shifted their venue preferences outside of their community due to the expansion of online ordering at supercenters in neighboring counties. However, the COVID-19 pandemic has had devastating economic consequences for many communities, as several stores on the cusp of closing their doors were either exacerbated or accelerated to do so. Taken together, this high poverty community faced many challenges during the time periods, which evidences itself through the store closure and changes in dietary intake. 

Although the community still has two remaining grocery stores, our findings indicate residents are traveling beyond the county to procure their groceries. Furthermore, although more women than men were represented in the study sample, this may provide more insight when investigating this research question, as women tend to be the food shoppers for households more frequently than men, thus proving more insightful information as to the grocery shopping and dietary habits of households [39]. This is consistent with previous research in another rural Appalachian county where the primary grocery store closed; as Miller et al. found, even when another grocery store was available in their community, residents traveled farther from home to complete a majority of their grocery shopping at a supercenter [19]. Although food access may have decreased with the grocery store closure, availability may have increased for those shoppers who switched from utilizing a grocery store to a supercenter more frequently. Although not assessed in this study, it is well documented that ultra-processed, calorically dense products with high sugar and fat content are readily available and contribute to diet-related chronic disease [40]. However, the shift in grocery shopping landscapes also provides increased opportunity for affordable and nutritious options. Data indicate slight increases in diet quality when those with limited access shop farther from home and have greater product selection [41]. This could explain the significant changes in dietary intake in part due to the change in availability of foods and beverages therefore impacting the purchases of consumers observed in the present study. Thus, traveling to reach venues with greater availability may positively influence dietary intakes, creating a unique opportunity to engage rural residents at these locations. 

Although fruit and vegetable intake improved between time point one and time point two among this cohort study group, unmeasured consequences such as longer travel time, less frequent shopping trips, and increased funds required for gas may result from this requirement to grocery shop at a store further away. Although this study did not explicitly measure these factors, insight can be gleaned to better understand the profound and lasting impacts of a store closure on a community. Therefore, food accessibility in rural areas remains a major concern that must be addressed. Policy makers should consider strategies to support rural grocery stores, particularly when operating in struggling communities or economically distressed regions of the U.S. Online grocery shopping in particular has steadily increased in the past three years, contributing USD 106 billion of the USD 1.04 trillion grocery market in 2019, and is expected to continue climbing [42]. Utilization among customers has increased, with 45% of consumers reporting doing more grocery shopping online than before the pandemic [43], and federal agencies have begun to try and match these spending habits. In 2020, the USDA expanded their online grocery shopping pilot program among SNAP recipients, in which SNAP enrollees could redeem their benefits for online grocery purchases at qualifying retailers [44]. This provided a great expansion to food access among this highly vulnerable low-income population in particular; thus, online grocery shopping should be considered among other federal policymakers and organizations as a viable economic effort to invest in that may potentially improve other dietary intake outcomes simultaneously. This is a potential modem to stimulate commerce in rural communities, which typically display higher rates of SNAP participation than urban areas [45]. 

This study has several limitations, as our study was conducted among a small sample size and had a 48.7% attrition rate between data collection time points at time point one and follow-up. Furthermore, the COVID-19 pandemic could have impacted shopping behaviors, and therefore results should be interpreted with caution in terms of their generalizability. Changes in the reported alcohol consumption were not probed or speculated on in this manuscript, as these behaviors are beyond the expertise of the study team, and this should be examined in future research. Survey data were collected via self-reported measures, and social desirability bias is possible. Additionally, since travel time was not collected at time point one and time point two data collection did not occur until after the grocery store closure, it is possible that travel times were naturally increased as a result of having decreased grocery options. Future research should be done to continue to investigate how multifaceted interventions could be implemented within these communities and food environments when economic and sociological conditions lead to impacts upon public health and chronic disease prevalence. 

## 5. Conclusions

This study contributes to the dire call to action of policy makers that direct incentivized activity of food outlets in rural communities, such as the Healthy Food Financing Initiative in urban environments, which directly supports the improvement of healthy food availability. Policymakers and federal agencies should consider the growing utilization and offering of online grocery shopping as a promising investment to both improve food access while economically supporting grocery outlets in struggling rural communities. This study reinforces this directive for future research, as our cohort population indicated significant changes in dietary consumption patterns when changing their venue for purchasing food and beverage items. As chronic disease prevalence, food insecurity, and obesity rates continue to pervasively exist among rural Appalachian communities, environmental and systems level of influence should be targeted as a method to more broadly impact and improve the nutritional habits of frequent shoppers. 

## Figures and Tables

**Figure 1 ijerph-19-03506-f001:**
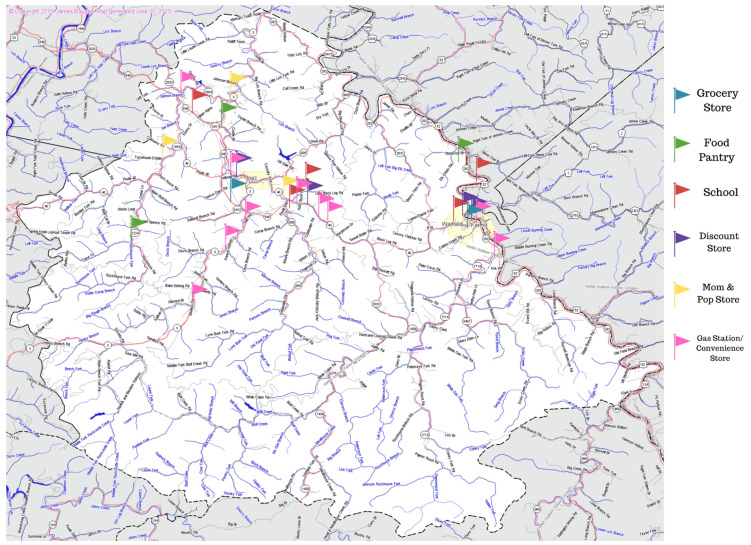
Map of Martin County, KY (229 sq. miles) with all available food outlets available to the residents within the county designated.

**Figure 2 ijerph-19-03506-f002:**
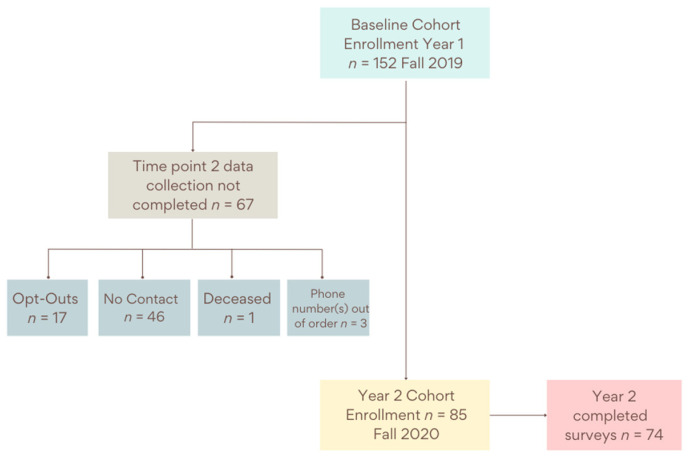
Time point two enrollment process from time point one among HOP project cohort study participants.

**Table 1 ijerph-19-03506-t001:** Demographic characteristics among the sampled cohort participants between time point one and time point two.

Demographic Characteristic	Time Point One *n* = 152 *n* (%)	Time Point Two *n* = 74 *n* (%)
Age (median (range), in years)	56.0 (22–84)	59.5 (22–85)
Gender		
Male	53 (34.9)	23 (31.1)
Female	99 (65.1)	51 (68.9)
Race		
White	150 (98.7)	74 (100)
Non-white	2 (1.3)	0 (0)
Education		
Less than high school	66 (43.4)	26 (35.1)
High school graduate	55 (36.2)	30 (40.5)
Post-high school	31 (20.4)	18 (24.3)
Household Income		
<$20,000	90 (60.4)	44 (61.1)
≥$20,000	59 (39.6)	28 (38.9)
SNAP Participation		
Yes	60 (39.5)	31 (41.9)
No	92 (60.5)	43 (58.1)
Travel time to store for grocery shopping		
Less than 10 min	-	19 (25.7)
10–20 min	-	14 (18.9)
20–30 min	-	24 (32.4)
More than 30 min	-	17 (23.0)
Where do you get most of your groceries? ^1^		
Grocery Store	122 (80.3)	48 (63.5)
Supercenter	31 (20.4)	36 (48.6)
Discount Store	1 (0.7)	2 (2.7)
What is the primary reason you shop there? ^1^		
Price	64 (42.1)	22 (29.7)
Location	62 (40.8)	31 (41.9)
Quality	12 (7.9)	11 (14.9)
Variety	14 (9.2)	19 (25.7)

^1^ Percentages may add to >100% due to multiple selections made by some participants.

**Table 2 ijerph-19-03506-t002:** Grocery shopping habits and changes from time point one to time point two among cohort participants.

		Time Point Two Shopping		
Time Point One Shopping		Yes *n* (%)	No *n* (%)	*p*-Value ^1^
Grocery Stores	Yes	42 (56.8)	14 (18.9)	0.04 *
	No	5 (6.8)	13 (17.6)	
Super Stores	Yes	16 (21.6)	3 (4.1)	0.0004 *
	No	20 (27.0)	35 (47.3)	
Discount Stores	Yes	0 (0.0)	1 (1.4)	0.56
	No	2 (2.7)	71 (95.6)	
Price Motivated	Yes	8 (10.8)	19 (25.7)	0.38
	No	14 (18.9)	33 (44.6)	
Location Motivated	Yes	19 (25.7)	13 (17.6)	0.84
	No	12 (16.2)	30 (40.5)	
Quality Motivated	Yes	4 (5.4)	2 (2.7)	0.10
	No	7 (9.5)	61 (82.4)	
Variety Motivated	Yes	6 (8.1)	3 (4.1)	0.01 *
	No	13 (17.6)	52 (70.3)	

^1^ McNemar’s tests were used to determine changes between time point one and time point two. * Indicates *p* < 0.05.

**Table 3 ijerph-19-03506-t003:** Food and beverage dietary intake changes from time point one to time point two among cohort sample that reported changing venues to grocery shop.

Food & Beverages	No Change	Already	Started	*p*-Value ^1^
Fruit Juice	−0.2 ± 0.6	−0.0 ± 0.2	0.1 ± 0.6	0.184
Fruit	−0.3 ± 0.8	−0.4 ± 0.8	0.2 ± 0.8	0.036 *
Tomato Sauce	−0.1 ± 0.2	0.0 ± 0.3	0.1 ± 0.3	0.035 *
Fruit/Vegetable overall	−0.7 ± 2.4	−0.7 ± 2.2	1.4 ± 2.7	0.008 *
Water (grams)	149.0 ± 650.2	0.9 ± 315.2	72.8 ± 684.7	0.675
Total Beverage (Calories)	−146.0 ± 387.8	12.7 ± 273.6	53.7 ± 208.3	0.057
Total Beverage (grams)	−229.0 ± 1331.9	−85.9 ± 569.3	−19.3 ± 811.3	0.756
SSB (calories)	−18.9 ± 248.0	29.3 ± 179.4	82.4 ± 214.0	0.272
SSB (grams)	−29.0 ± 633.4	47.7 ± 444.0	164.4 ± 722.1	0.539
Milk (grams)	−150.9 ± 355.6	−13.3 ± 190.6	−47.5 ± 313.5	0.204
Alcohol (grams)	−6.1 ± 20.2	−2.7 ± 11.7	17.3 ± 54.1	0.036 *

^1^ ANOVA tests used to determine dietary changes between time point one and time point two. * Indicates *p* < 0.05.

## Data Availability

The data presented in this study are available on request from the corresponding author. The data are not publicly available due to privacy and ethical protection of study participants.

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
