# Peer review of "The Impact of a Grocery Store Closure in One Rural Highly Obese Appalachian Community on Shopping Behavior and Dietary Intake"

_ijerph, 2022, doi:10.3390/ijerph19063506_

Round 1

Reviewer 1 Report

The following paper evaluates the effect of a closure of a grocery store
on food outlet shopping choice, dietary intake, and purchasing patterns among a rural and low-income population in Eastern Kentucky (USA).

The manuscript describes an important aspect of how a specific environmental change can impact the dietary and food shopping choices, which eventually can significantly affect the overall health.

The paper described the methodology and results in a very detailed and clear way. Please find below my few comments and suggestions related to the article. 

Abstract: I suggest removing the enumeration of the sentences (e.g., 1, 2,...) and keeping the abstract as one paragraph.

Introduction: line 67: the terms "natural experiment" are not very familiar.

Materials and methods: Overall, the methodology is clearly explained. Adding the map was an excellent way to illustrate the study setting. However, the inclusion criteria of the participants as well as the sample size calculation are missing. 

Analysis: Were the grocery shopping habits also assessed for normality?

Results: I suggest moving the lines 178-182 to the "discussion" paragraph as the "results" paragraph should not include any interpretations. 

Discussion: The food and beverage dietary intake changes were not discussed enough. Please elaborate further. In addition, the limited number of participants should be added to the limitations of the study.

Reviewer 2 Report

Thank you for allowing me to review this contribution entitled

The impact of a grocery store closure in one rural highly obese Appalachian community on shopping behavior and dietary intake

In general, the topic of research and the approach of the study are interesting.

There is consistency among research title, the research question/goal and contribution/answers for research questions but I suggest necessary changes to incorporate in the document for publication

Materials and methods

Line 139 onwards

Information/justification should be included as to why only fruit and vegetable and beverage consumption was used to assess dietary intake.  

Results

L195  L200  It is necessary to cross-check the % because in some cases they do not coincide with those shown in table 1.

L214 It seems a comment more appropriate for the discussion section than for the results.

Discussion

L289 I don't think it can be said that  dietary intake moderately improved.   If anything, reference could be made to improved consumption of fruit and vegetables.

In line with the comment on methodology, intake of other food groups is not discussed. Although the work focuses on fruit, vegetables and beverages, it would be relevant to discuss the possible access to a wide variety of ultra-processed products or products with high sugar or saturated fat content such as breakfast cereals, pastries, snacks, etc. which are not discussed and could also be available in large variety in supermarkets. 

Furthermore, there is no mention or consideration of the increase in alcohol consumption, which in any case would not be an improvement. 

It is important that the discussion includes these considerations and expands on the limitations. 

Round 2

Reviewer 2 Report

The authors have collected the different proposals for change suggested.
However, it is necessary to revise the text from line 200 onwards as in line 203 they indicate "Further, when focusing only on the time point two cohort..."  but from L200 to L203 the data also correspond only to the time point two. 

Author Response

Reviewer 2,

Thank you for catching this discrepancy. We have modified the sentence to reflect more clearly that data from time point two is being presented. The line now reads, "Further, time point two descriptive statistics revealed..."

Thank you for the opportunity to revise and improve this manuscript. 

Rachel Gillespie, on behalf of all co-authors